# Benchmarking Offline Reinforcement Learning Algorithms for E-Commerce Order Fraud Evaluation

**Soysal Degirmenci, Chris Jones**
Amazon
soysald@amazon.com

## Abstract

Amazon and other e-commerce sites must employ mechanisms to protect their millions of customers from fraud, such as unauthorized use of credit cards. One such mechanism is order fraud evaluation, where systems evaluate orders for fraud risk, and either "pass" the order, or take an action to mitigate high risk. Order fraud evaluation systems typically use binary classification models that distinguish fraudulent and legitimate orders, to assess risk and take action. We seek to devise a system that considers both financial losses of fraud and long-term customer satisfaction, which may be impaired when incorrect actions are applied to legitimate customers. We propose that taking actions to optimize long-term impact can be formulated as a Reinforcement Learning (RL) problem. Standard RL methods require online interaction with an environment to learn, but this is not desirable in high-stakes applications like order fraud evaluation. Offline RL algorithms learn from logged data collected from the environment, without the need for online interaction, making them suitable for our use case. We show that offline RL methods outperform traditional binary classification solutions in SimStore, a simplified e-commerce simulation that incorporates order fraud risk. We also propose a novel approach to training offline RL policies that adds a new loss term during training, to better align policy exploration with taking correct actions.

## 1 Introduction

Millions of customers shop at Amazon and other e-commerce sites every day. A fraction of these customers are bad actors who attempt to use stolen payment instruments (e.g., credit cards) or compromised accounts to place unauthorized orders. To protect customers from bad actors, e-commerce sites employ mechanisms to identify and deter bad actors, and ensure that only authorized orders are completed. One such mechanism is order fraud evaluation: when a customer places an order, a system assesses risk of payment fraud and may "pass" the order, or cancel the order and suspend (aka "fraud") the account. These systems typically employ binary classification machine learning solutions to assess risk, where fraud risk models are trained on features and outcomes (labels) of historical orders.

We seek to ensure that customers are satisfied in the long-term. For order fraud evaluation, this implies a system that considers both the immediate financial losses of fraud, such as the chargeback loss to reimburse victims, and the long-term impact of incorrectly cancelled orders and frauded accounts. For example, when legitimate customers' accounts are incorrectly frauded, they may need to endure a remediation process to reinstate accounts, preventing purchases during the reinstatement process, and frustrating and discouraging customers.[1]

---

[1]Note that abandonment of the reinstate process by a legitimate customer leads to incorrect labels, since such fraud actions are presumed to be correct.

Offline Reinforcement Learning Workshop at Neural Information Processing Systems (NeurIPS 2022), New Orleans, Louisiana.

Taking actions to optimize long-term impact can be formulated as a Reinforcement Learning (RL) problem. In recent years, deep RL algorithms have proven successful in many fields [19, 8, 10]. Standard versions of these methods require online interaction with an environment in order to explore relationships between actions and return to learn optimal policies. However, such online interactions are inadvisable in high-stakes applications such as our use case, where errors during exploration and learning can lead to significant negative impact on customers. Offline RL [7, 15] algorithms learn from logged data, collected from the environment over a period of time. These methods do not require online interaction with the environment to learn optimal policies.

In this work, as a step towards using offline RL policies for order fraud evaluation:

- We introduce the SimStore simulation package, a simplified e-commerce simulation where customers (including bad actors) place orders that are evaluated by a risk policy, which selects an action: Pass or Fraud.

- We formulate order fraud evaluation as an RL problem. An episode (i.e., a sequence of events until termination) consists of a sequence of orders and order evaluations of a customer.

- We generate multiple levels of logged data using SimStore, for the purpose of training policies. The levels, medium, and expert, are determined by the success of the online policy used to collect the data.

- We propose a novel approach to training offline RL policies, suitable to applications such as order fraud evaluation. Specifically, we introduce a new loss during training to train policies that maximize long-term returns and short-term risk jointly.

- We compare several offline RL algorithms to a binary classification method, based on the various levels of logged data. We show that some of the offline RL algorithms, including our proposed novel method, are either on-par or outperform the other methods.

## 2 Related Work

Reinforcement Learning has been an active field for many years, with recent successful results in a wide range of domains [19, 8, 10]. Standard approaches in RL typically use online learning. In online RL, methods require interaction with an environment to update policy parameters, evaluate the performance of the updated policy, for many times until the algorithm terminates. However, this requirement may be unfeasible for some domains due to cost (e.g. autonomous driving, robotics) and safety (e.g. healthcare, fraud evaluation). Offline RL algorithms have been proposed to alleviate these problems. In offline RL, a policy is learned from logged data, collected from an environment over a period time, interaction with the environment is not required. The policy used affects the data distribution collected from an environment. When a policy is learned using an offline dataset, the data distribution when the learned policy is in use differs from the logged data, resulting in a data distribution shift. This remains as the fundemantal problem with offline RL and several different approaches have been proposed to tackle it.

Offline RL methods can be grouped into two categories in terms of learning and utilizing a model of the environment. Model-based offline RL methods [9, 11, 27, 26] train a model of the environment using state-action transitions from the logged data. These methods utilize the learned model to generate synthetic episodes, controlled by the policy being trained. The policy parameters are updated using a combination of real episodes (from the logged data) and synthetic ones until convergence. On the other hand, model-free methods [6, 7, 14, 24] learn a policy that maps states to actions to maximize returns directly.

Fraud detection and prevention using algorithmic methods is a crucial area for many industries because lack of it may cause significant financial losses and negative experiences on customers. Typically, they are posed as a supervised learning problem with fraud-related variables as inputs and fraud labels as outputs [21, 1]. Recently, researchers have proposed use of RL for fraud detection and prevention [28, 23, 17]. In [28], the authors aim to improve an e-commerce platform's impression allocation and minimize seller fraudulent behavior jointly using online RL, whereas in [17], the authors propose a method to learn the behavior of a bad actor agent, whose goal is to maximize financial losses.

# 3   SimStore

SimStore is a stochastic, simulated, OpenAI Gym [2] compatible environment that mimics an online e-commerce store subject to order fraud risk. Many of the behaviors simulated in SimStore are governed by random processes, drawn from pre-configured probability distributions (see Appendix A). SimStore is comprised of an inventory, where a set of product listings (including their identifier, category and price) are generated randomly based on configuration settings. An initial set of customers is generated during initialization, and new customers are generated throughout simulation. Customers come in three types:

- Regular Customer: Regular customers do not intend to commit fraud. When their accounts are incorrectly frauded, they may choose to reinstate to continue shopping or abandon, based on a configured, random variable.
- Bad Actor, Sleeper Attack: The bad actor engaged in a sleeper attack places non-fraudulent, low-priced orders for a period of time, and then places fraudulent, high-priced orders.
- Bad Actor, Immediate Attack: The bad actor engaged in an immediate attack only places fraudulent, high-priced orders.

While bad actors follow prescribed attack patterns, the realization of each attack is governed by pre-configured probability distributions, as described in Appendix A. Figure 1 presents order evaluation workflow used in SimStore for a customer:

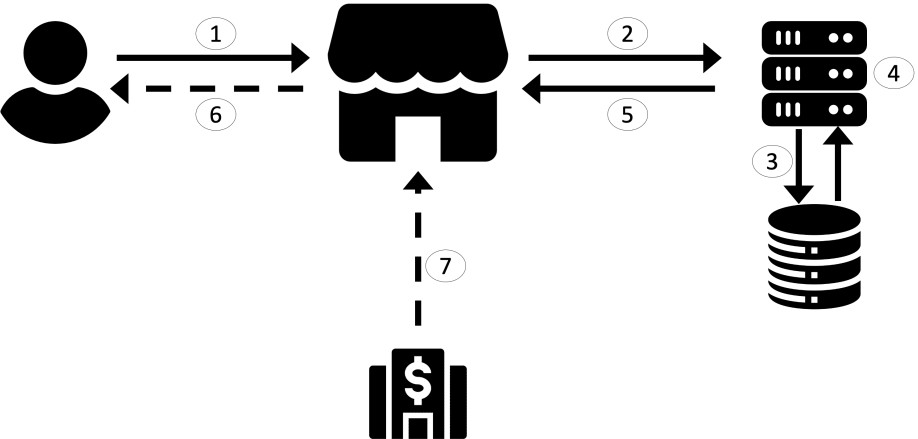

Figure 1: Order evaluation workflow.

1. Customer selects an item from the store and places an order. The fraudulent intent of this order is decided by the customer at this stage.
2. Store receives the order and routes it to order fraud risk evaluation policy.
3. Order fraud risk evaluation policy generates features. Some of the features are generated from the order metadata, whereas some are generated from historical data (i.e., past customer spending).
4. The features are used to evaluate fraud risk and decide on an action.
5. The action (Pass or Fraud) is returned to the store.
6. If the action is Pass, the order is fulfilled. If the intent of a passed order is non-fraudulent, it is added as revenue to the records. If the action is Fraud, the order is not fulfilled and the customer account is suspended. (Customer accounts of regular customers may choose to reinstate to continue shopping whereas accounts controlled by bad actors may not reinstate.)
7. If the action is Pass and the intent of the order is fraudulent, the store receives a chargeback from bank. This is added as a chargeback loss to the records.

## 4 Problem Statement

Order fraud evaluation is formulated as an RL problem where an agent, a fraud risk evaluation policy, interacts with its environment, SimStore and the orders placed by customers. The environment is assumed to be a partially observable Markov decision process (POMDP) where the agent receives a sequence of observations $o_i$ at times $t_i$, takes actions $a_i$ using a policy $\pi(a|o)$. Then, the agent receives a reward $r_i$ and an inferred binary outcome $\hat{y}_i \in \{0, 1\}$ for each state it has evaluated and acted on. The inferred binary outcome is the inferred fraudulent intent of the order placed.

In this application, the origin of the sequences $S$ received by the agent from the environment is orders placed by customers in chronological order. This sequence consists of a set of subsequences, $\mathbb{S} = \cup_{j=1}^{C} S_j$, where a subsequence $S_j$ corresponds to order evaluations for orders placed by the customer $c_j$, in chronological order. Therefore, a subsequence corresponds to an episode.

Placement of an order triggers a fraud risk evaluation. SimStore assigns the fraud intent of the order at origination, the true binary outcome $y_i$ ($= 1$ implying a fradulent intent). True binary outcomes are never available to policies for training. True and inferred binary outcomes can be different, depending on the action taken. Specifically, when a system takes a Fraud action on a non-fraudulent order and the customer abandons the account, the inferred outcome $\hat{y}_i$ is different from the true outcome (1 vs. 0).

During fraud risk evaluation, the system generates a set of risk variables $o_i$ for the order placed by customer $c_i$. These variables are derived from the the order itself and historical orders that preceded it. All algorithms presented in this paper use the same set of variables. For example, a variable may count the total number of orders a customer has placed in their lifetime[2].

After risk variable generation, the policy $\pi(a|o)$ takes an action $a_i \in \{0, 1\}$, where 0 represents the action Pass, and 1 represents the action Fraud. When the action is Fraud, the order is cancelled, and the account of customer $c_i$ is closed, preventing the customer from placing further orders until the account is reinstated (only available to regular customers). The action taken on the order affects future orders of the customer of interest. When Fraud action is taken on an order by a regular customer, they may choose to reinstate or abandon their account, stopping placing orders.

The agent receives a reward $r_i$, and an inferred binary outcome $\hat{y}_i$ after an action is taken. In this work, the reward is designed to be the financial impact of the action, i.e. either an increase or decrease in revenue. When the order is non-fraudulent and passed, it is equal to the order amount (a positive number that contributes to the revenue). When the order is fraudulent and passed, the result is a chargeback, a financial loss incurred by the agent, and is equal to the negative of the order amount. Moreover, when the order is frauded, the reward is equal to 0 (since there is no revenue gained or financial loss incurred). This choice of reward is designed to optimize for the long-term customer satisfaction (i.e., the revenue, as satisfied customers continue to spend on SimStore) at the expense of short-term financial losses incurred by the store (i.e., the chargebacks, as passed fraudulent orders are reimbursed). In reality, the reward signal is not received immediately in cases of payment risk; it may take days, weeks or months before a chargeback loss is received, since the victim must identify the fraudulent charge and engage with the bank to complete the chargeback process. Furthermore, some orders that are passed and labeled as fraudulent ($\hat{y}_i = 1$) do not result in any chargebacks. We leave incorporating these features into SimStore as future work.

The goal of the agent is to maximize the cumulative reward over a distribution of sequences, named returns $R_i$. Different from the typical definition of $R_i$, for an observation $o_i$ whose order was placed by customer $c_i$, we only account for the future rewards related to the customer $c_i$. Another difference is that we incorporate a temporal discount factor; instead of assuming a fixed interval time-step between orders in a sequence, the contributions of future rewards in the return are based on their time differences. Specifically, we calculate return values as $R_i = \sum_{j=i}^{\infty} I(c_j = c_i) r_j \gamma^{t_j - t_i}$, where $I(\cdot)$ is the indicator function and is equal to 1 when $c_j = c_i$ (i.e., the orders $i$ and $j$ are placed by the same customer) and is 0 otherwise, and $\gamma \in [0, 1]$ is the discount factor. This return $R_i$ is the time-discounted future spending and chargeback losses expected from a customer, at a given point in time.

The goal is to find an optimal policy for the POMDP $M$ that maximizes the expected performance when it starts from an initial state distribution $\rho_0$. The optimization problem of interest can be written

---

[2]For the full list of variables used, please see Appendix A.2.

as $\max_\pi \mathbb{E}_{\rho_0}(\pi, M) = \mathbb{E}_{o \sim \rho_0} \left[ \sum_{i=0}^{\infty} R_i | o_0 = o \right] = \mathbb{E}_{\tau \sim p_\pi(\tau)} \left[ \sum_{k=0}^{T-1} R_k(\tau) \right]$, where $p_\pi(\tau)$ is the trajectory $\tau$ distribution induced by policy $\pi$. This optimization can be done in various ways by online RL methods. In offline RL, there is no interaction with the environment during policy optimization. A dataset $D = \{(o_i, t_i, a_i, c_i, r_i, \hat{y}_i, o_i')\}|_{i=0}^{|D|}$ [3] is collected by a family of policies and the goal is to learn an optimal policy by leveraging this dataset only, without any interaction with the environment.

# 5 Experimental Setup

## 5.1 Data Collection

To evaluate various offline RL methods, we first collect logged data from SimStore based on varied types of policies. An important factor in the success of offline RL methods is their ability to generalize the out-of-data distribution, i.e., state-action pairs that are not present in the logged data available during training. Thus, it is standard practice [5] to evaluate offline RL algorithms using several levels of quality: medium and expert. In contrast to standard offline RL datasets used to benchmark methods, we do not include a dataset collected by using a policy that takes random actions, since exploitation of such a policy by bad actors would be catastrophic for a business. More information about data collection can be found in Appendix B.

## 5.2 Offline RL Algorithms

Each dataset is used to train several types of algorithms as described below. For all the cases, we incorporate a mechanism that closes the accounts of customers whose orders are passed and result in chargebacks. This is inspired by how order fraud evaluation systems function in practice, to stop accumulation of chargeback losses from accounts used by bad actors as soon as possible. We evaluate and compare the following algorithms: Behavioral Cloning (BC), Binary Gradient Boosted Trees (BGBT), Deep Q-Network (DQN) [18], our novel method, Multi-Objective DQN (MODQN), Batch-Constrained deep Q-learning (BCQ) [6, 7], Critic Regularized Regression (CRR) [24], and Conservative Q-Learning (CQL) [14]. More information about these algorithms is presented in Appendix C.

### 5.2.1 Multi-Objective Deep Q-Network (MODQN)

Deep Q-Networks [18] aim to learn an optimal state-action function $Q_\theta^*(o, a)$ parameterized by $\theta$, that estimates the expected return at state $o$ when action $a$ is taken. The parameters $\theta$ are typically the weights of a neural network with states $o$ as inputs and actions $a \in A$ as outputs. For a logged dataset $D$, this becomes the minimization problem:

$$\min_\theta \sum_{k=0}^{|D|} \left( \hat{R}_{\theta'}(r_k, o_k, D, \gamma) - Q_\theta(o, a) \right)^2, \tag{1}$$

where $\hat{R}_{\theta'}(r_k, o_k, \gamma) = r_k + \gamma \max_a Q_{\theta'}(o_k', a)$ is the expected return and uses the Bellman equation. Note that the parameters of the two Q-networks, $\theta$ and $\theta'$ are different. $\theta'$ is updated to be equal to $\theta$ on a cadence defined by a hyperparameter[4], which was shown to improve the performance of the Q-networks empirically. Once the Q-network is trained, the policy $\pi(a|o) = arg\,max_a Q_\theta(o, a)$ is used to take actions.

To better align the loss function of the DQN to our use case, we propose to add a term that uses the inferred outcome of each record. For some $Q_\theta(o, a)$, we first compute its softmax over $a$. The softmax outputs can be interpreted as probability values associated with each action. Therefore, we add a binary cross entropy term that incentivizes these probabilities to be close to inferred outcomes.

---

[3] $o_i'$ represents the state variables received after action $a_i$ is taken. Due to our definition of return, this corresponds to the next order evaluation of the customer $c_i$.

[4] This is called target update frequency in hyperparameters.

Table 1: Normalized net revenue for different datasets and algorithms

| Algorithm
Dataset | BC | BGBT | DQN | MODQN | BCQ | CRR | CQL |
|---|---|---|---|---|---|---|---|
| medium | 87.75
± 2.74 | 68.26
± 12.28 | **89.88**
± 8.04 | 83.06
± 5.50 | 81.61
± 3.11 | 76.07
± 5.38 | 83.56
± 10.07 |
| expert | 93.37
± 4.11 | 93.68
± 3 | 93.66
± 4.21 | **96.11**
± 2.25 | 92.34
± 4.27 | 92.82
± 3.77 | 94.76
± 1.99 |

The new objective function is:

$$\min_{\theta} \sum_{k=0}^{|D|} \left( \hat{R}_{\theta'}(r_k, o_k, D, \gamma) - Q_\theta(o, a)) \right)^2 + \beta \sum_{k=0}^{|D|} (\hat{y}_i \log q_\theta(o, a = 1) + (1 - \hat{y}_i) \log q_\theta(o, a = 0)),$$

(2)

where $q_\theta(o,:)$ is the values when softmax operation is applied to $Q_\theta(o,:)$, $q_\theta(o,a) = e^{Q_\theta(o,a)} / \sum_{j=1}^{A} e^{Q_\theta(o,a_j)}$. We call this method Multi-Objective DQN (MODQN).

## 6   Results

During evaluation, we collect several types of metrics, and use the net revenue as the primary success metric as it quantifies long-term customer satisfaction (through spending) and short-term financial losses (through chargebacks). Table 1 shows the performance of the implemented algorithms using different types of data. Each policy is trained by using the logged data once. Then, the trained policy interacts with several SimStore environments that share the same set of hyperparameters but are generated from different random seeds. The table presents the normalized mean and standard deviations. The net revenue of each run is normalized so that it is between 0 (no revenue) and 100 (the best scenario). The net revenue is 0, based on a risk policy that Frauds all orders. The best net revenue is scaled to 100, based on an oracle risk policy that uses true outcomes $y_i$ to take actions.

The results show that several offline RL algorithms outperform the approach based on a binary classifier. Deep Q-network-based methods (including our proposed method, MODQN) generally outperform other methods, but we don't observe that any offline RL methods outperform others by a large margin. Except for the CRR with medium-level dataset, the results indicate that there exists a set of hyperparameters that performs well for each algorithm after an extensive hyperparameter search. (For more details on how the hyperparameter search is conducted, please see Appendix D.)

## 7   Conclusion

We proposed to formulate order fraud evaluation for e-commerce as a reinforcement learning problem to maximize long-term customer satisfaction. We introduced SimStore, a simplified e-commerce simulation, and showed that offline RL methods outperform the traditional binary classification method in terms of net revenue. We also introduced a novel way to train offline RL methods by adding a loss term during training that trades off between short-term losses (i.e., passing fraudulent orders) and long-term gains (i.e., avoiding incorrect closure of legitimate accounts).

We showed that the RL methods, including the novel approach proposed, are superior to the approach based on binary classification, for SimStore in our selected configuration. We hypothesize that this is because the RL methods attempt to directly maximize the return (or at least, a version that discounts future rewards), while binary classification relies on distinguishing fraudulent orders, which is subject to label bias due to customers who choose not to reinstate.

While offline RL is promising, there remains future work. We plan to evaluate other offline RL algorithms, especially model-based methods, as a part of our benchmark suite. In order to make SimStore more realistic, we plan to add more features such as delayed reward mechanism, more types of bad actors, and bad actors that learn from past experience (i.e., bad actors as RL agents). We will explore other configurations of SimStore, such as when reinstatement is near-certain, and label bias less of a factor. We are also interested in exploring how results change as dataset size

grows. Finally, we performed hyperparameter selection for all the methods (including the binary classification method) based on how the trained policies performed in the simulated environment. Hyperparameter selection without needing to interact with the environment is a known phenomenon with offline RL [19], which we also plan to pursue.

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

# A  Appendix A: SimStore

## A.1  SimStore hyperparameters

SimStore includes an environment wrapper compatible with OpenAI Gym [2], a popular API for training RL algorithms. This allows rapid exploration of many RL packages, under a variety of configurations of the store and its customers.

Behaviors simulated in SimStore are governed by random processes drawn from probability distributions. This section presents the set of hyper-parameters.

### A.1.1  High-level hyperparameters

- Number of customers: Number of customers at the beginning of the simulation.
- Ratio of regular customers: Expected ratio of good customers at the beginning of the simulation and during new customer sign-up.
- Ratio of bad actors, immediate attack: Expected ratio of bad actors that follow the immediate attack strategy to all bad actors. Used at the beginning of the simulation.
- Mean time between new customer sign-ups: The time between new Customer sign-ups is modeled to follow an exponential distribution. This parameter is set separately for regular customers and bad actors.
- Number of products.
- Number of product categories.

### A.1.2 Customer-level hyperparameters

- Maximum time between orders placed: The time between two consecutive orders a customer places orders is modeled as an exponential distribution. This is set separately for regular customers and each bad actor type.

Regular Customer:

- Mean time between consecutive orders: This is drawn from an exponential distribution.
- Probability of reinstate: Probability of a regular customer being reinstated if they are incorrectly Frauded. This is drawn from a Bernoulli distribution.
- Mean time between consecutive orders, post-reinstate: This is drawn from an exponential distribution.

Bad Actor:

- Percentile of most expensive items to target: This determines the subset of items Bad Actors will attempt to target, based on item price. Then, the bad actor randomly chooses an item randomly from this subset.

Bad Actor, Sleeper Attack:

- Mean number of orders before attack: This is modeled as a Poisson distribution with the mean as a hyperparameter.
- Mean time between consecutive orders before attack: This is drawn from an exponential distribution.
- Probability of chargeback before attack: Probability of an order being chargeback before attack. This is drawn from a Bernoulli distribution.
- Probability of chargeback during attack: Probability of an order being chargeback during attack. This is drawn from a Bernoulli distribution.
- Mean time between consecutive orders during attack: This is drawn from an exponential distribution.

Bad Actor, Immediate Attack:

- Mean time between consecutive orders: This is drawn from an exponential distribution.
- Probability of chargeback: Probability of an order being chargeback. This is drawn from a Bernoulli distribution.

### A.1.3 Inventory-level hyperparameters

- Price mean and standard deviation: Price of each item is drawn from log normal distribution.

Table 2 presents the hyperparameters used in the experiments.

### A.2 Variable list for Order Fraud Evaluation

Table 3 presents the list of variables computed and used for order fraud evaluation in SimStore. Most of the variables are generated by using historical evaluations. The variables payment_method_risk and location_risk are generated when the order is generated. These variables are modeled to represent the risk associated with the payment method, physical and digital location related to the order. They are modeled as Beta distributions whose parameters are different based on the fraudulent intent of the order; they are more likely to have larger values when the order is fraudulent. Figure 2 shows the empirical distribution of both variables used in the experiments.

## B   Data Collection

For each case, we run the simulation for three months, collecting two levels of logged data. The levels are determined by how successful each policy is, where the success metric is net revenue, i.e. the

Table 2: Hyperparameters used in SimStore

| Name | Value |
|------|-------|
| Number of customers | 1000 |
| Ratio of regular customers | 0.8 |
| Ratio of sleeper attack bad actors among bad actors | 0.2 |
| Mean time between new customer sign-ups, regular customer | 4 hours |
| Mean time between new customer sign-ups, bad actors | 15 minutes |
| Regular customer, mean time between consecutive orders | 2 days |
| Regular customer, probability of reinstate | 0.2 |
| Regular customer, mean time between consecutive orders, post-reinstate | 4 days |
| Bad actor, percentile of most expensive items to target during attack | 10 |
| Bad actor, sleeper attack, percentile of cheapest items to target before attack | 10 |
| Bad actor, sleeper attack, mean time between consecutive orders before attack | 2 days |
| Bad actor, sleeper attack, mean number of orders before attack | 5 |
| Bad actor, sleeper attack, probability of chargeback before attack | 0.01 |
| Bad actor, sleeper attack, probability of chargeback during attack | 0.99 |
| Bad actor, sleeper attack, mean time between consecutive orders during attack | 6 hours |
| Bad actor, immediate attack, probability of chargeback | 0.99 |
| Bad actor, immediate attack, mean time between consecutive orders | 6 hours |
| Item price, mean | 50 |
| Item price, standard deviation | 100 |

Table 3: Variables used for Order Fraud Evaluation

| Name |
|------|
| customer_past_lifetime_num_orders |
| customer_past_lifetime_dollars_spent |
| customer_past_lifetime_num_unique_categories |
| customer_past_lifetime_num_unique_asins |
| customer_days_since_first_order |
| customer_days_since_last_order |
| customer_past_lifetime_num_chargeback_orders |
| customer_past_lifetime_dollars_spent |
| order_total_price |
| item_past_lifetime_num_orders |
| item_past_lifetime_num_unique_customers |
| payment_method_risk |
| location_risk |

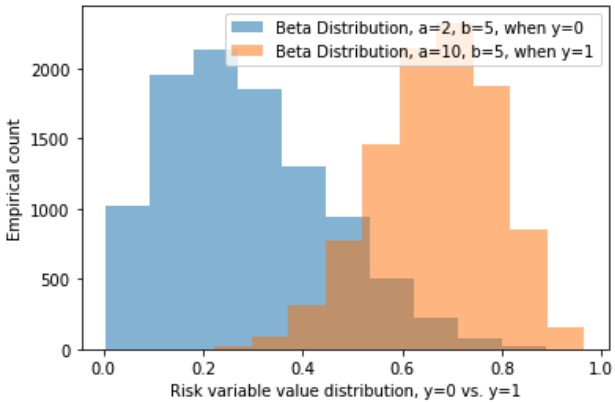

Figure 2: Empirical distribution of payment method and location risk variables.

cumulative order amounts of passed, non-fraudulent orders minus passed, fraudulent orders (resulting in chargebacks), during the entire run. The two levels of data are:

- Medium: To mimic the real-world data collection, we devise a policy based on an ensemble binary classification models, using gradient-boosted decision trees. Starting with a random policy (with probability $0.9$ of passing an order), the binary classifier is re-trained every day during simulation. It uses the variables as inputs and inferred binary outcomes as outputs. The data is split into train and test, and it is trained to classify fraudulent orders until the ROC-AUC of the test dataset does not improve anymore. More details of the algorithm can be found in Section C.2. Hyperparameters are chosen so that the policy used to collect this dataset performs "medium" level, compared to the best case scenario where there are no wrong decisions.

- Expert: The same methodology used to collect Medium data collection is used, with different hyperparameters.

## C  Offline RL Algorithms

### C.1  Behavioral Cloning (BC)

A policy trained with behavior cloning learns to imitate the policies used during data collection. Formally, the policy is learned by minimizing $\sum_{l=0}^{|D|} L(\pi(: |o_l), a_l)$ with respect to $\pi$, where $(o_l, a_l)$ is the data element collected, and $L$ is a loss function that quantifies the discrepancy between two terms. We model $\pi$ as a neural network, and use binary cross entropy as the loss function.

### C.2  Binary Gradient Boosted Trees (BGBT)

In order to mimic typical order fraud evaluations systems, we train a policy based on an ensemble of binary classification models. Each model is a decision tree, trained by using gradient-boosting with target labels $\hat{y}$, the inferred binary outcomes. For a given record $o_i$, an ensemble of trees produces a prediction probability $f_{GBT}(o_i)$. The policy passes orders when $f_{GBT}(o_i) \leq \epsilon$ and frauds otherwise, where $\epsilon$ is a selected threshold. After the model is trained, this threshold is chosen by maximizing a metric of interest in the validation dataset, where the choices are presented in Appendix D.2.

### C.3  Batch-Constrained deep Q-learning (BCQ)

Batch constrained deep Q-learning is an offline RL method [6, 7] where the algorithm for discrete action space can be thought as a variant of Q-learning. With a behavior policy $\pi_\beta(a|o)$ that mimics the policy used to collect logged data, a parameter ($\tau$ in the original paper, $0 \leq \tau \leq 1$) can be chosen to adjust for the learned policy to take actions. Smaller values of this parameter would result in a typical Q-learning setting whereas larger values would imitate the behavior policy, the policy used to collect logged data.

### C.4  Critic Regularized Regression (CRR)

Critic Regularized Regression [24] is an offline RL algorithm that can be regarded as a selective behavioral cloning based on a learned critic. The critic is a Q-network and the policy is encouraged to mimic the behavior policy used to collect the data when the estimated returns are larger.

### C.5  Conservative Q-Learning (CQL)

Conservative Q-Learning [14] is an offline RL algorithm proposed to address the distributional shift between the dataset used to train policies and learned policies themselves. This shift causes an overestimation in Q-learning, and the algorithm uses additional regularization terms during training to estimate conservative Q values. Following the formulation in [13], the minimization problem of

CQL is:

$$\min_\theta \sum_{k=0}^{|D|} \left( \hat{R}_{\theta'}(r_k, o_k, D, \gamma) - Q_\theta(o, a)) \right)^2 + \lambda \sum_{k=0}^{|D|} \left( \log \sum_{a'} \exp(Q_\theta(o_k, a')) - Q_\theta(o_k, a_k) \right).$$

(3)

Similar to DQN, the policy that takes the action with the maximum estimated return $Q_\theta(o, a)$ is used to take actions.

## D  Offline RL Algorithm hyperparameter search

For each algorithm, we identified a set of hyperparameters to tune to maximize the performance during evaluation. For each case, we allocated a budget of $128$ evaluations. These $128$ evaluations are performed on randomly selected $128$ sets of parameters from the superset defined using random search. We used Ray Tune [16] as the framework to conduct these experiments. For each run, we used the logged data (simstore-medium or simstore-expert) collected from SimStore to train a policy. Then, we ran each policy on SimStore with the same hyperparameters five times with different random seeds.

For the algorithm(s) that use neural networks, we tried different normalization methods for state variables $s$ and rewards $r$. For state variables, we evaluated two variants, one transformation where each variable is between $0$ and $1$ (named minmax), and another transformation with subtracting the mean and dividing by the standard deviation (named standard). The rewards were transformed so that they are always between $-1$ and $1$. We implemented the algorithms starting from Tianshou library [25]. Tianshou uses PyTorch [20] in the backend. We modified the implemented algorithms so that there is a time component in logged data, and the return is computed based on our formulation as described in Section 4.

For each method, the available dataset is organized into episodes per customer. Then, these episodes are split into training and test ($75\%$ and $25\%$ respectively) sets. For BGBT, this was done on an order level. During training, loss of test set is recorded, and the model snapshot whose test loss is the minimum is returned as the best policy. Early stopping was performed for all cases if the loss of test set did not improve for 50 epochs (50 trees for BGBT). We used Adam [12] as the optimization method for all experiments. All neural networks use ReLU as the activation function in intermediate layers.

For all cases that use discount factor $gamma$ as a parameter, we use the following discrete space $[0.1, 0.2, 0.3, 0.4, 0.5, 0.6, 0.7, 0.8, 0.9, 0.95, 0.99]$. For all cases with Q-networks that update the target network, we use a parameter called target update frequency. The discrete space used for this parameter is $[50, 100, 250, 500, 1000, 2500, 5000]$.

### D.1  Behavioral Cloning (BC)

Table 4 shows the hyperparameter search space and the best parameters found for both datasets.

Table 4: BC, hyperparameter search space and best parameters

| Name | Range | simstore-medium | simstore-expert |
|------|-------|-----------------|-----------------|
| batch size | $[64, 128, 256]$ | 64 | 64 |
| learning rate | $(1e-4, 1e-1)$, log uniform | $2.19e-3$ | $2.2e-3$ |
| number of layers | $[2, 3, 4, 5]$ | 5 | 4 |
| max number of epochs | $[50, 100, 250]$ | 50 | 100 |
| layer size | $[32, 64, 128]$ | 128 | 64 |
| state transformation | $[standard, minmax]$ | $standard$ | $minmax$ |

### D.2  Binary Gradient Boosted Trees (BGBT)

For BGBT, we use the XGBoost Python package [3] to train the models.

Table 5 shows the parameters used based on the hyperparameter search (the parameters not presented here are the default parameters specified by the package). The early stopping scheme tracks ROC-AUC of the test set. After the ensemble is trained, we choose the threshold where the probabilities larger than it takes Fraud action, and Pass action otherwise. The threshold chosen maximizes the F1-score or the reward on test set.

Table 5: BGBT, hyperparameter search space and best parameters

| Name | Range | simstore-medium | simstore-expert |
|---|---|---|---|
| max number of trees | $[500, 1000, 1500]$ | 1500 | 1000 |
| max depth | $[3, 4, 5, 6, 7, 8]$ | 3 | 8 |
| learning rate | $(1e-3, 1e-1)$, log uniform | $2.79e-3$ | $7.8e-2$ |
| colsample by tree | $(0.25, 1.0)$, uniform | 0.36 | 0.93 |
| colsample by level | $(0.25, 1.0)$, uniform | 0.71 | 0.28 |
| subsample | $(0.25, 1.0)$, uniform | 0.67 | 0.51 |
| scale pos weight | $(1e-1, 1e2)$, log uniform | 0.48 | 0.15 |
| threshold choice metric | $[f1, reward]$ | $reward$ | $reward$ |

### D.3 Deep Q-Networks (DQN)

Table 6 presents the results for DQN. For Q-networks, we train an ensemble of networks where the number of ensembles is denoted as number of ensembles in the table below. Each neural network in the ensemble is initialized randomly, and the mean of the outputs is used training and inference. Using double Q-networks [22] is also a parameter in our search. Finally, we also vary the number of steps to compute the approximate return (the equation to compute 1-step return is shown in Section **??**), this parameter is denoted as number of estimation step.

Table 6: DQN, hyperparameter search space and best parameters

| Name | Range | simstore-medium | simstore-expert |
|---|---|---|---|
| gamma | (see Appendix D) | 0.9 | 0.3 |
| batch size | $[64, 128, 256]$ | 128 | 128 |
| learning rate | $(1e-4, 1e-1)$, log uniform | $6.7e-2$ | $5.4e-3$ |
| target update frequency | (see Appendix D) | 250 | 50 |
| number of layers | $[2, 3, 4, 5]$ | 4 | 4 |
| number of ensembles | $[2, 4, 8, 16]$ | 8 | 16 |
| number of estimation step | $[1, 2, 3, 4]$ | 4 | 3 |
| max number of epochs | $[50, 100, 250]$ | 250 | 50 |
| layer size | $[32, 64, 128]$ | 32 | 32 |
| is double | $[False, True]$ | $True$ | $True$ |
| state transformation | $[standard, minmax]$ | $standard$ | $minmax$ |

### D.4 Multi-Objective DQN (MODQN)

Table 7 presents the results for MODQN. Compared to DQN hyperparameter search, the only additional hyperparameter is $\beta$, the weight of the additional term introduced during training.

### D.5 Batch-Constrained deep Q-learning (BCQ)

Table 8 presents the results for BCQ. The parameter that controls the epsilon-greedy noise added in evaluation is denoted as eval eps while the threshold ($\tau$ in Equation 19 in [6]) for unlikely actions is denoted as unlikely act threshold. The regularization weight for imitation logits is represented as imitation logits penalty.

Table 7: MODQN, hyperparameter search space and best parameters

| Name | Range | simstore-medium | simstore-expert |
|---|---|---|---|
| gamma | (see Appendix D) | 0.6 | 0.8 |
| batch size | $[64, 128, 256]$ | 64 | 256 |
| learning rate | $(1e-4, 1e-1)$, log uniform | $6e-3$ | $8.7e-4$ |
| target update frequency | (see Appendix D) | 5000 | 50 |
| number of layers | $[2, 3, 4, 5]$ | 5 | 5 |
| number of ensembles | $[2, 4, 8, 16]$ | 8 | 8 |
| number of estimation step | $[1, 2, 3, 4]$ | 3 | 3 |
| max number of epochs | $[50, 100, 250]$ | 100 | 50 |
| layer size | $[32, 64, 128]$ | 64 | 64 |
| is double | $[False, True]$ | $True$ | $True$ |
| state transformation | $[standard, minmax]$ | $minmax$ | $minmax$ |
| beta | $(1e-4, 1e1)$, log uniform | $7.3e-2$ | $7.9e-3$ |

Table 8: BCQ, hyperparameter search space and best parameters

| Name | Range | simstore-medium | simstore-expert |
|---|---|---|---|
| gamma | (see Appendix D) | 0.7 | 0.9 |
| batch size | $[64, 128, 256]$ | 64 | 128 |
| learning rate | $(1e-4, 1e-1)$, log uniform | $5.1e-4$ | $3.9e-2$ |
| target update frequency | (see Appendix D) | 250 | 500 |
| number of layers | $[2, 3, 4, 5]$ | 3 | 2 |
| number of estimation step | $[1, 2, 3, 4]$ | 2 | 1 |
| max number of epochs | $[50, 100, 250]$ | 100 | 50 |
| layer size | $[32, 64, 128]$ | 128 | 64 |
| is double | $[False, True]$ | $True$ | $True$ |
| eval eps | $(0.0, 0.99)$, uniform | $9.4e-3$ | 0.29 |
| unlikely act threshold | $(0.0, 0.99)$, uniform | 0.86 | 0.77 |
| imitation logits penalty | $(1e-4, 1e1)$, log uniform | $3.2e-3$ | 0.91 |
| state transformation | $[standard, minmax]$ | $minmax$ | $standard$ |

## D.6 Critic Regularized Regression (CRR)

Table 9 presents the results for CRR. The type of the function used ($f$ in the paper) is represented as policy improvement mode. The parameters used when policy improvement mode is $exp$ are denoted as beta and ratio upper bound.

Table 9: CRR, hyperparameter search space and best parameters

| Name | Range | simstore-medium | simstore-expert |
|---|---|---|---|
| gamma | (see Appendix D) | 0.99 | 0.4 |
| batch size | $[64, 128, 256]$ | 128 | 256 |
| learning rate | $(1e-4, 1e-1)$, log uniform | $1.6e-3$ | $5.3e-3$ |
| target update frequency | (see Appendix D) | 50 | 250 |
| number of layers | $[2, 3, 4, 5]$ | 4 | 4 |
| max number of epochs | $[50, 100, 250]$ | 100 | 50 |
| layer size | $[32, 64, 128]$ | 64 | 32 |
| policy improvement mode | $[binary, exp, all]$ | binary | all |
| beta | $(1e-3, 1e1)$, log uniform | N/A | N/A |
| ratio upper bound | $(1e-3, 1e2)$, log uniform | N/A | N/A |
| state transformation | $[standard, minmax]$ | $standard$ | $minmax$ |

## D.7 Conservative Q-Learning (CQL)

Table 10 the results for CQL. For CQL, we followed the Tianshou implementation and used Quantile Regression Deep Q-Networks (QR-DQN) [4] to train the networks. Number of quantiles used in the Q-network is denoted as number of quantiles. During inference, we use the average of outputs to choose an action. The weight of the regularizer ($\lambda$ in Equation 3) is denoted as lambda.

Table 10: CQL, hyperparameter search space and best parameters

| Name | Range | simstore-medium | simstore-expert |
|---|---|---|---|
| gamma | (see Appendix D) | 0.8 | 0.9 |
| batch size | $[64, 128, 256]$ | 256 | 64 |
| learning rate | $(1e-4, 1e-1)$, log uniform | $3.5e-2$ | $1.4e-4$ |
| target update frequency | (see Appendix D) | 500 | 250 |
| number of layers | $[2, 3, 4, 5]$ | 4 | 4 |
| number of quantiles | $[2, 4, 8, 16]$ | 16 | 4 |
| number of estimation step | $[1, 2, 3, 4]$ | 4 | 2 |
| max number of epochs | $[50, 100, 250]$ | 100 | 50 |
| layer size | $[32, 64, 128]$ | 32 | 32 |
| state transformation | $[standard, minmax]$ | $standard$ | $minmax$ |
| lambda | $(1e-3, 1e1)$, log uniform | 0.79 | $2.2e-3$ |

