# OpenReview forum: "Benchmarking Offline Reinforcement Learning Algorithms for E-Commerce Order Fraud Evaluation"
_NeurIPS.cc/2022/Workshop/Offline_RL — Offline RL Workshop NeurIPS 2022_

### Official Review · Reviewer_zd17 · 2022-10-19
**Structure and contributions need work**

**Rating:** 3
**Confidence:** 4

**Review:**

This paper describes the issue of fraud in e-commerce and motivates the use of RL as a way of detecting fraud, taking actions that trade off termination of suspected fraud with disruption or customer dissatisfaction. The authors have developed an OpenAI Gym-compatible environment that simulates this problem. The paper is well-written in terms of clarity and flow. However, much of the most useful parts of the paper are in the appendix. Not just the details of the simulator but also the details of how experiments were conducted and all the algorithms used. I didn't notice that their method was only described in the appendix until I went through it. There is also no discussion of related work. I would say the lack of development of their own method in the body of the text is reason alone to reject this paper for now.

In general, this paper is missing contributions to the topic of offline RL. They successfully motivate how RL and offline RL could potentially be good approaches for fraud mitigation and model the problem in a way that makes this feasible. However, in the end they simply apply different offline RL algorithms to a simple simulated problem. Their novel method is not introduced in the body of the paper, and ultimately is only a small adjustment suited to this particular problem. I would like to see the motivation for and development of their method at the forefront of the paper contributions if this paper were to be admitted into this workshop in another form. I would like to see how it mitigates some of the common pitfalls of offline RL approaches.

---

### Official Review · Reviewer_LNCQ · 2022-10-20
**Interesting application, but have questions on whether modeling task as an MDP overly complicates things.**

**Rating:** 6
**Confidence:** 4

**Review:**

The authors of this paper apply offline RL algorithms to the task of fraud detection in e-commerce. Though prior solutions typically learn a binary classifier and apply it independently on each order, the authors argue that offline RL is needed to maximize long-term revenue over a sequence of orders.

Overall, the paper covers an interesting application that is explained well. They also consider a sufficiently large set of offline RL algorithms to benchmark on this new environment. However, I am not convinced that this problem requires an MDP to model it. Namely, to my knowledge, the actions that the agent takes (deeming an order as good or fraudulent) does not appear to affect future orders. Since there is no transition model underlying the SimStore environment, I wonder if bandit policies are sufficient to solve the task. If what I said is incorrect, then it would be helpful to explicitly state what the transition dynamics of the MDP would be in Section 3, as the authors currently only elaborate on the reward function.

Nevertheless, I still think it would be helpful to compare to myopic/bandit policies closer to the ones considered in prior literature, just to observe if using RL leads to noticeable improvement. As an analogous task, recommender systems should also aim to satisfy long-term satisfaction, and therefore can be solved using RL. However, a majority of literature, and practical applications, primarily consider bandit policies because they are simpler and work "well enough" already. Hence, I am unsure if using RL on this fraud detection task will also be overly complicated.